# Clinical Provider Perspectives on Remote Spirometry and mHealth for COPD

**DOI:** 10.3390/nursrep15110402

**Published:** 2025-11-15

**Authors:** Susan McCabe, Jessica Madiraca, Lianne Cole, Emily Morgan, Terri Fowler, Whitney Smith, Catherine O’Connor Durham, Kathleen Lindell, Sarah Miller

**Affiliations:** 1College of Nursing, Medical University of South Carolina, Charleston, SC 29466, USA; mccabesu@musc.edu (S.M.); morgaemi@musc.edu (E.M.); fowlerte@musc.edu (T.F.); smithwhit@musc.edu (W.S.); durhamc@musc.edu (C.O.D.); lindellk@musc.edu (K.L.); 2School of Nursing, University of Delaware, Newark, DE 19716, USA; jessrn@udel.edu; 3Boston Children’s Hospital, Boston, MA 02115, USA; lianne.cole@childrens.harvard.edu

**Keywords:** remote spirometry, mobile health, digital health, smartphone application, remote patient monitoring, healthcare providers, chronic obstructive pulmonary disease, qualitative research, implementation science

## Abstract

**Background/Objectives:** COPD is a leading cause of death in the US, with higher morbidity and mortality in rural areas that lack specialized pulmonary care. Mobile health (mHealth) tools, including remote spirometry, may fill this gap, yet healthcare provider (HCP) perspectives on utility and implementation of remote spirometry and mHealth for COPD management in these settings remain unexplored. This study aimed to examine HCPs’ perspectives of mHealth with remote spirometry to inform future implementation in rural and low medical access settings. **Methods:** Five HCPs working in rural or medically limited settings in South Carolina participated in a deliberative discussion focus group. A semi-structured interview guide was used to explore insights into feasibility, clinical utility, and implementation needs. Transcripts were analyzed using thematic analysis to identify facilitators, barriers, and implementation considerations. **Results:** Participants reported that remote spirometry and mHealth had potential to support COPD treatment, increase healthcare access, and support self-management. Key facilitators identified were access to smartphones, potential for individualized COPD care, and visual tools for patient engagement. Barriers included risk of time and workload burden, data information overload, and technological limitations. Participants emphasized the need for team training, thoughtful integration into existing workflows, customizable data displays, and robust support for patient onboarding. **Conclusions:** Providers viewed mHealth applications with remote spirometry as a valuable tool with potential to improve COPD care but identified critical implementation needs. Participants emphasized that successful integration would require structured support, user-centered design, and attention to reimbursement and workflow alignment to enhance sustainability and patient/provider engagement.

## 1. Introduction

Chronic obstructive pulmonary disease (COPD), the 5th leading cause of death in the United States, presents an economic burden of $31.3 billion annually, projected to increase to $60.6 billion by 2029 [1,2,3]. The burden of COPD is higher in rural and medically underserved areas (regions with provider shortages or limited healthcare access) [4], where patients face higher risks for complications and mortality due to lack of specialized pulmonary care and limited healthcare access [2,3,5,6,7,8,9]. Due to a shortage of pulmonologists and limited availability of pulmonary rehabilitation centers in rural areas, primary care providers are often responsible for the early detection and ongoing management of COPD [5,6,7,10]. This is inconsistent with guidelines-based standards and places significant strain on providers due to complexity of case management, resource limitations, and fragmented care coordination [9].

Many clinics in medically limited settings, including free clinics and federally qualified health centers (FQHC), do not have spirometry equipment for testing, which is the “gold standard” biomarker for COPD diagnosis and monitoring disease progression [11,12]. Spirometry is essential for distinguishing COPD from other respiratory or cardiovascular conditions, staging disease severity, and assessing therapeutic response. Objective measurement of forced expiratory volume in one second (FEV1) and forced vital capacity (FVC) allows the clinician tailor treatment plans according to evidence-based guidelines outlined by the Global Initiative for Chronic Obstructive Lung Disease (GOLD). Patients are typically referred to pulmonary function testing labs, where they face long wait times for appointments and encounter travel and access barriers. As a result, providers must rely on clinical judgement for COPD diagnosis and treatment plans, which are often based upon presumptive data such as the physical exam, symptom reports, or self-reports of past diagnoses from other providers [5,6,7,10]. Because declines in FEV1 are associated with increased exacerbation risk, hospitalization, and mortality, the inability to conduct regular spirometry introduces a critical gap in longitudinal disease monitoring and can contribute to poorer outcomes and greater healthcare utilization.

The gap in specialized pulmonary care in remote settings can be addressed by home-based mobile health (mHealth) solutions [7,13,14,15]. Rapid advances in technology have led to integration of medical equipment with mobile devices, remote patient monitoring (RPM) options, and mHealth apps [16], offering numerous accessible mHealth solutions to providers and patients. Digital mHealth COPD interventions may improve adherence to treatment [17,18,19,20,21,22], exercise capacity [18,19,20,21,22,23], and early identification of exacerbation symptoms [18,19,20,23,24,25,26,27]. Among existing mHealth technologies, portable spirometers provide an affordable, easy-to-use RPM option for ongoing monitoring of lung function. This allows the patient to visualize changes in air flow when experiencing onset of dyspnea symptoms, even when pulmonary lab appointment availability, travel distances, and times are a barrier (frequent issue for those residing in rural areas) [28]. Expanding access to spirometry testing, through portable and remote spirometry devices, can address diagnostic and treatment gaps and improve long-term management of COPD in rural and medically underserved settings where resources are limited.

While patients and healthcare providers (HCP) are generally positive towards mHealth interventions for COPD, other studies have reported barriers such as technical issues, privacy or confidentiality concerns, limited patient eligibility, or challenges with patient motivation and engagement [16,29]. However, to our knowledge, no studies have specifically explored provider perspectives and preferences for using remote spirometry for patient monitoring in COPD, especially those residing in rural or medically limited settings. Therefore, the purpose of this study was to investigate provider perspectives on the utility, accessibility, and implementation of remote spirometry and mHealth for COPD in rural and low medical access settings (i.e., healthcare environments, such as free clinics and FQHCs, that primarily serve low-income and uninsured populations).

## 2. Materials and Methods

This was a health services initiative with two distinct phases from formative insights to applied testing. The first formative phase, reported here, was an investigation of provider preferences, including barriers and facilitators, of implementation of remote spirometry for remote patient monitoring of patients with COPD. Results from this will be used to inform the second implementation phase of this initiative, which will build upon provider perspectives gathered in this project by integrating mHealth tools and remote spirometry in rural and low medical access settings. This phase will focus on feasibility, usability, and workflow integration, while exploring patient preferences and acceptance. Anticipated outcomes include identifying barriers and facilitators to adoption and refining training and support strategies to inform larger-scale implementation efforts.

### 2.1. Study Design

Providers working in rural or medically limited settings (free clinics and federally qualified health centers) in South Carolina participated in a deliberative discussion focus group [30] (qualitative descriptive framework). This design was selected to generate robust discussion with interaction between group members. As deliberative discussion focus groups occur within a social context, they provide an effective tool for research and allow an opportunity to discuss pros and cons in a discussion, increasing the utility of focus group data. Participants were recruited using convenience and snowball sampling methods through professional and clinical networks to identify providers currently delivering care in settings that serve rural and low-income populations in South Carolina. South Carolina ranks among the lowest nationally in access to specialized pulmonary care and contains a large number of federally designated rural and medically underserved areas. This regional focus allowed us to capture provider experiences that reflect their unique healthcare delivery challenges.

Participation in the deliberative discussion was voluntary, and participants could decline involvement or withdraw at any time. No personal identifiers were collected, and findings are reported in aggregate to maintain confidentiality. The interview script was developed by the principal investigator (PI) and co-investigator (co-I), who is a clinical provider, then subsequently refined by a senior qualitative methodologist. The semi-structured interview guide included open-ended or probing questions to prompt discussion on mHealth, technology in clinical practice, and care of COPD patients in medically limited settings. Example questions included, “Tell me about your previous experiences with mHealth,” “What mHealth features may be most helpful for your COPD patients?” and “What are your biggest concerns about integrating remote spirometry or an mHealth app into your practice?”

### 2.2. Procedure

The deliberative discussion focus group was conducted in person in a comfortable, private meeting room at an academic medical center central to all participants. The discussion lasted approximately one hour. Expert moderation was central to the deliberative format, and the session was facilitated by a senior PhD Nurse Scientist and a PhD student who is also a practicing Nurse Practitioner. Both facilitators had formal training and prior experience in qualitative research methods, including conducting focus groups and applying deliberative discussion techniques. Before the session, the facilitators reviewed key principles of reflexivity and neutrality to ensure balanced participation and promote open deliberation amongst participants. One provider participated synchronously and virtually via Microsoft Teams due to illness. The focus group was recorded via Microsoft Teams software (Version 25255.703.3981.5698) and hand transcribed.

### 2.3. Data Analysis

A qualitative descriptive approach was used to analyze the data. Focus group recordings were transcribed verbatim, de-identified, and stored securely [31]. Thematic inductive analysis was conducted inductively using an iterative process [32,33]. A junior nurse scientist (SMc) led initial coding, carefully identifying and organizing meaningful segments of data. Codes were then reviewed and refined collaboratively with a nurse scientist (SM) and qualitative researcher (JM) to ensure consistency, accuracy, and reliability. Discrepancies in coding were discussed and resolved through consensus. Themes were refined through an iterative process and finalized based on their coherence and relevance to the dataset. Because data were de-identified and collected as part of the quality improvement initiative, transcripts could not be returned to participants. Instead, analytic rigor and confirmability were supported by consensus coding and iterative review by multiple researchers.

## 3. Results

A total of five healthcare providers, including doctorally prepared nurse practitioners (n = 3), a pharmacist, and a pharmacy student, participated in this study. Participants all had a minimum of 15 years clinical practice experience with a range of 15–23 years of experience. All participants were practicing in medically limited settings in South Carolina, including free clinics (n = 3) and FQHCs (n = 2), and all had experience providing care in rural communities. Some sites were located in small rural towns that functioned as regional hugs, and others were suburban free clinics serving as critical access points for individuals without insurance who would otherwise have limited or no access to healthcare. The providers represented the following specialties and fields: family practice, adult-gerontology, ambulatory primary care, and internal medicine. All providers contributed to free clinic services in settings that serve uninsured and underinsured patients. Across the settings, providers cared for predominantly low-income populations with a high proportion of uninsured and Medicaid-insured individuals, racially and ethnically diverse patient groups (including Black, White, and Hispanic communities).

### 3.1. Themes

The analysis of the focus group data highlighted seven themes related to the utilization of remote spirometry and mHealth for COPD management in a medically limited setting. Themes were organized and classified into facilitators (3), barriers (3), and implementation needs (1). The study findings are discussed in Table 1, with the qualitative findings presented in a narrative format with verbatim extracts to support the seven themes.

### 3.2. Facilitators

Providers expressed overall positive perceptions of the use of mobile health technology in clinical practice. The three themes that emerged as facilitators were as follows: (1) the potential to address barriers in access to care, (2) COPD care and the ability to pinpoint more accurate pulmonary etiology, and (3) the prospect of facilitating COPD self-management for their patients.

(a)Addressing barriers in access to care: All participants discussed the need for equitable access for their patients and identified mobile health technology as a potential solution.


*“In addition to the limited resources, our patients often have limited capacity, whether that’s transportation or whatever it may be, that they don’t. Even if we can get the referrals and the appointments set up and all of that for all the specialists, they often can’t make that.”*
(Participant 5)


*“But we do work with populations that have a lot of barriers, right, a lot of social determinants that are working against them and one of those is very, very, low health literacy. So, I would anticipate that that would be a huge challenge …”*
(Participant 4)

Researchers inquired about the accessibility and availability of technology resources needed for mHealth technology. Participants who had worked with mHealth or telehealth services in the past in this setting reported that mobile phone availability and access was not a concern.


*“We haven’t gotten too many barriers to where we surprised actually at how (pause) willing … to meet or used to telehealth technologies, and so forth. I think that’s becoming less and less of a barrier. And most quite, I mean, it’s very rare that they don’t have a smartphone now.”*
(Participant 4)


*“… they do have that smartphone. So, it would be … is that a usable form or are they gonna need the bigger screen? Because I do think if they can use their phone most of them could.”*
(Participant 2)

(b)COPD care: Several participants discussed how mHealth may be a solution to address challenges in caring for patients with COPD in this setting, due to the complexity of COPD. Providers reported they often had to treat patients based upon presumptive data, due to barriers (financial, physical proximity, long waitlist) in obtaining spirometry testing. Providers discussed how this kind of technology could bridge this gap, which would address their frustrations and challenges of not having available diagnostic testing available in the clinic.


*“Constrictive … restrictive, mixed. What are we looking at, you know? Are we missing asthma combination with COPD? Do we have a problem? I mean we’re just going through the clinical problem solving to determine what.”*
(Participant 4)


*“… it’s a real struggle to try to predict which one it is. It’s hard for us clinically to make the right decisions when we don’t have the right information.”*
(Participant 4)


*“And this technology would at least help us quickly rule out most pulmonary etiologies and help us triage because our patients have very limited resources.”*
(Participant 5)

Under the COPD care theme, participants noted that mHealth and RPM tools may facilitate evidence-based treatment and management. This included the potential to remotely monitor therapeutic medication changes in real-time instead of waiting for follow-up appointments.


*“For the COPD patient, it’s really the true value is diagnostic and then changing meds and seeing if you can get an improvement.”*
(Participate 4)


*If there was a way that “… they could maybe even just log like they did their long acting or they needed to use their breakthrough. Like that we could see trends on that.”*
(Participant 1)


*“Just the value of it. Right. So, more information means better decision making when it comes to problem solving.”*
(Participant 4)

(c)COPD self-management: Participants felt that the use of mHealth and remote spirometry could be useful for the self-management of COPD. This could be accomplished by educating and empowering patients to become more knowledgeable about their disease process and monitor their disease progression. The ability to view lung function in a visual graphic was cited as a benefit of remote spirometry and mHealth in general.


*“… from the mHealth perspective, it’s also really because sometimes when you want to do something different with the patients’ treatment plan, you want to change a medicine or go about something different. They may be resistant, and they have in their own brain what this should look like or why … then when you say, “No, look here. Let’s look at your sugars and your trends,” and you can show them in real time, then it makes the conversation much easier. And I think for the patient, then they understand the rationale behind it.”*
(Participant 5)

### 3.3. Barriers

Three themes identified as barriers emerged from the data. The first theme focused on time and workload concerns, the second addressed issues of data volume and information overload, and the third related to technical or technology-related challenges.

(a)Time and workload: Time concerns included the amount of time required to adopt and implement mHealth or remote spirometry, and additional time burden that might be placed on the provider in terms of workload.


*“I don’t know in the middle of seeing a lot of patients—like that’s just a lot and that’s probably not going to happen. So, I think as a provider to really have meaningful integration, there has to be support in that space for the provider”*
(Participant 5)


*“You can kind of see the general feasibility and I think especially for our patient population like. Umm. Is this really feasible or are we like, yeah, this is way too much work.”*
(Participant 5)

(b)Data volume: One key barrier expressed by all participants was the potential for information overload resulting from the volume of data generated by mHealth technologies. Concerns included how to manage and interpret data, where data would be stored, and the timing and frequency of data delivery. Participants expressed a desire for integration of the data for meaningful clinical interpretation.


*“It’s got to be reliable and accurate. And something that I can understand, umm, because otherwise it’s just too much work and you don’t have time for all that.”*
(Participant 5)


*“You’d want to be able to see both of those. You want to be able to narrow it down. You want the summary stuff to just kind of be eye-popping, you know are there any major warnings or concerns.”*
(Participant 4)

(c)Technology barriers: Along with data volume, concerns for technology barriers and quality of healthcare interaction were expressed by participants. One anticipated technological barrier was the risk of malfunction with the use of technology, including unreliable internet. Participants spoke of their past experiences using telehealth or mobile health technology. They reported that they had experiences with patients not being fully engaged with the provider during some of these interactions, often due to other distractions in the home or remote environment.


*“The other for me in doing telehealth was always, the patient’s there and they’re engaged but there’s a lot of distractions.”*
(Participant 2)


*“even the Internet going in and out. That was the other thing. That sometimes we would have a bad connection …. If I was going to have glitchy things, it’s either that they were in their car and like really, really preoccupied.”*
(Participant 2)


*“like they’re in a car with eight other people … it can be very difficult to have a good, meaningful conversation, especially something that’s requiring some thought and action because there’s just a lot going on.”*
(Participant 5)


*“it can be really distracting to like carrying on an actual visit because you do not have their undivided attention and there’s no thought to giving you. It’s just like you’re the girlfriend calling …. I found it as a provider very challenging at times.”*
(Participant 5)

### 3.4. Implementation Needs

The final theme, implementation needs, included desire for clear processes, data interpretation, and data delivery. Participants reported a need for clear guidelines on the use of remote spirometry, including guidelines and instructions for patients, as well as guidelines for provider interpretation and utilization. There is a need for a clear process of how devices would be used for maintenance, monitoring of acute exacerbations, and timing of spirometry measurements.


*“Are you gonna come up with, kind of like, “Here’s some generalized guidelines that how it might be beneficial and how to use it and when to use it?”*
(Participant 4)


*“And as a provider, this would be really helpful for me, is there any literature out there on at-home spirometry. … How to use it? When to use it? … Frequency? Exacerbations versus just control and maintenance? Umm. And, if so, as a provider that would be really helpful to know because then that would help us when determining when to use it.”*
(Participant 4)

Participants also expressed a strong willingness and desire for training, technological support, and implementation guidance. Key implementation needs included support with the technology itself, particularly during the onboarding process, facilitating setup, and troubleshooting issues. Participants also emphasized the need for easy-to-use training and support resources, both for themselves and their clinical teams. This included resources for educating students during clinical rotations (thus alleviating the education burden from the provider), clear communication on managing reports and data output, and training for patients and clinical staff.


*“[for] a provider to really have meaningful integration, there has to be support in that space for the provider: of getting the patient set up, getting them educated, getting them as well as myself connected, and then it has to be reliable- Like the reports coming in, the information coming in.”*
(Participant 5)


*“We’ve learned with mHealth…the support has- the technology support has to be there and the follow-up, if you really want it to happen. Because if the patient gets home and they’re like, “Wait, what is this thing?” Or “… it’s not working right.” Nine times out of 10, they are not going to contact you to tell you that …”*
(Participant 5)


*“… the easier the training, the quicker, and the more folks that can do the training like a student, or an MA, or … Like that’s the critical stuff for that’s really important for it to actually happen in real life practice.”*
(Participant 5)


*“if you did have an on-site ability to do the education like before that patient left with the device for instance. And you had a facilitator that enabled them to connect, and you go through the process. That actually works fairly well.”*
(Participant 4)

Expanding upon data output and management, the discussion provided insight into preferred data displays. This included how data is displayed to the provider, with providers preferring grouping into colors for ease of interpretation and efficient clinical management. Reports should clearly distinguish values and categorize them if possible.


*“We’re in a hurry, so colors, things that allow you to quickly look at something and interpret. Mainly, like just to look and say, ‘Oh good. If I want to come back and dig down into that later that’s fine.’”*
(Participant 5)


*“If for instance you have your FEV1 versus your FVC … and you’re categorizing like your COPD patient … it actually would be nice to just be like, ‘Oh, they have moderate- it just categorizes it right there, versus, “What does it mean?’” You know, I got to go back and look at…”*
(Participant 4)


*“Critical results. I think we want …. Like in, real, more real time.”*
(Participant 1)

When prompted to reflect on preferences for data delivery (email, text, in-chart alert), summary data vs. detailed insight, and a visual overview on a dashboard, participants discussed a desire for flexibility on data display. They expressed wanting the ability to customize data presentation based on individual preferences.


*“You’d want to be able to see both of those. You want to be able to narrow it down. You want the summary stuff to just kind of be eye-popping, you know are there any major warnings or concerns. And then, you’d want to dive down to see historical data as well. How does it compare? what’s happening over time.”*
(Participant 4)


*“Email’s fine or even a text message to my cell phone would be fine too or like a prompt to go, ‘Hey look at your, you know, you’ve got results.’”*
(Participant 1)

Finally, support was needed for navigating insurance and reimbursement of RPM services.


*“I don’t know what reimbursement is for this technology just because in our patient population this has not been—we haven’t had the ability to access it. So, it’s not been a thing. But I do not think but I do think in settings that can bill, that is a very important piece.”*
(Participant 5)

## 4. Discussion

This study explored healthcare provider (HCP) perspectives on the use of remote spirometry and mHealth tools for COPD management within rural and medically limited settings serving primarily uninsured patients. Facilitators, barriers, and implementation needs (Figure 1) were identified. For successful implementation, there is a need for qualitative investigation of stakeholder perception of implementation outcomes (acceptability, feasibility, and appropriateness) [34,35]. Findings from this study support that while clinicians are positive about remote spirometry, they face very practical hurdles in implementing into daily care. This is consistent with recent digital health studies identifying barriers in real-world implementation. Sustained telehealth success depends heavily on organizational readiness, training, and integration into existing systems [36,37,38]. Similarly, our participants identified the need for structured workflows, data displays optimized for clinical decision-making, and reliable technical support are priorities for successful long-term adoption. There is both promise and complexity of implementing mHealth interventions, including remote pulmonary monitoring, in clinical settings where systemic barriers to specialty care persist. Acceptance of an innovation does not guarantee its practical application in a busy, resource-constrained clinical environment.

Providers expressed optimism about the potential of mHealth-facilitated COPD care to enhance access, improve diagnostic accuracy, and support patient self-management. However, they simultaneously highlighted potential barriers and critical implementation considerations necessary for real-world integration. Specifically, workflow demands, data management, and the need for structured implementation support were identified by providers. This aligns with existing research on digital health in medically underserved settings, which emphasizes that effective adoption depends on incorporating clinical expertise into the design and implementation of interventions [39,40].

Also consistent with existing literature, participants perceived mHealth interventions as having the potential to extend pulmonary assessment capabilities to patients otherwise unable to access specialty services [14,15,16,22,29,41,42,43]. Participants underscored how geographic, financial, and transportation barriers impede timely access to spirometry, a diagnostic cornerstone for COPD, resulting in frequent reliance on presumptive clinical judgment or outdated records. Remote spirometry was viewed as a mechanism to “fill the gap,” providing actionable physiologic data to guide diagnosis, assess medication response, and inform management decisions. This finding aligns with prior work demonstrating that remote pulmonary monitoring may reduce exacerbation-related hospitalizations and improve disease self-management [16,24,29]. Participants also reported that when supported by robust workflows, mHealth tools could enhance clinical decision-making by enabling proactive care. The capacity to track medication use, monitor lung function trends, and visualize symptom fluctuations was perceived to facilitate more individualized, evidence-based care. Participants also identified remote spirometry as a potential patient engagement tool. By offering visual feedback on disease progression, providers felt this could help reinforce adherence and support shared decision-making, especially in populations with lower health literacy.

However, providers also shared reservations regarding the feasibility of implementing mHealth into routine practice. Foremost among the concerns was the risk of increased workload without the appropriate supportive infrastructure. Participants emphasized the need for integrated clinical workflows, technology support, and designated personnel to manage onboarding, data interpretation, and patient troubleshooting. Standardized protocols, particularly for differentiating between maintenance monitoring and acute exacerbation response, will be necessary. Providers also indicated that facilitating insurance reimbursement for remote physiologic monitoring (RPM) services will be critical for scaling these technologies in all settings. Without these supports, providers feared that remote spirometry data would become “just more work,” highlighting the critical importance of ensuring that digital health solutions alleviate, not exacerbate, provider burden. Studies have shown that poorly integrated digital tools can contribute to clinician burnout by increasing administrative workload and cognitive load, particularly when data are not actionable or embedded within existing systems [44,45].

The potential for information overload was also a salient theme. Participants articulated a clear preference for data outputs that are actionable, concise, and embedded within existing systems. They advocated for dashboard-style summaries using color-coded risk categories and the ability to drill down into historical trends only when clinically indicated. This reflects emerging evidence suggesting that optimizing the visual structure, functional alignment, and usability of digital health interfaces can enhance clinician engagement and reduce cognitive burden [46,47]. Design strategies should prioritize usability and workflow integration for improved user satisfaction and clinical decision-making [48]. This highlights the importance of user-centered design in mHealth tool development and suggests that digital health platforms should allow flexible customization of data displays based on provider roles and preferences [49]. Providers identified training needs for themselves, clinical support staff, and students rotating through clinics, supporting that sustainability of mHealth programs hinges on comprehensive training and ongoing technical support.

At the patient level, there were concerns about potential technology barriers such as unreliable internet access, limited technical support, and limited patient engagement during virtual visits. Participants shared prior experiences, including poor connectivity, distracting environments, and a lack of follow-through when technical issues occurred. Although smartphone access was not considered a major barrier, consistent with national data showing 88% of adults in rural areas own smartphones [50], the practical utility of these devices in clinical care remains limited in resource-constrained settings. While smartphone access is increasingly widespread, even in rural areas [50,51], access alone does not guarantee effective use of mHealth tools.

These findings support that successful uptake of remote spirometry in rural and medically limited settings will provide benefits to both the patient and the provider, but will require careful attention to organizational readiness, workflow integration, and reimbursement logistics. This includes leveraging existing staff roles, embedding digital tools into current workflows, and prioritizing low-bandwidth, user-friendly technologies. Equally important are targeted efforts to build digital literacy and foster patient engagement and self-management.

Overall, this study contributes to the growing body of research on the use of remote monitoring in chronic disease management by highlighting the clinical realities of providers working in rural and medically limited communities. This offers practical insight into the technological, operational, and educational supports needed to embed mHealth interventions in these and other primary care environments. Importantly, the deliberative discussion method allowed for a rich, interactive exploration of implementation facilitators and barriers, strengthening the ecological validity of the findings.

### Strengths and Limitations

To our knowledge, this is the first exploration of provider perspectives and preferences of the integration of remote spirometry with mHealth for COPD in rural and medically limited settings. A central strength is its provider-led approach, positioning clinicians as co-designers of digital health solutions tailored to the realities of resource-constrained environments. Unlike individual interviews, the deliberative discussion format fostered dynamic, peer-to-peer dialogue, encouraging participants to build on one another’s insights and clarify assumptions in real-time [30]. This approach generated rich, actionable findings, such as training needs, workflow integration, and data display preferences, that can inform future intervention design and scale-up.

This study has several limitations. First, a potential limitation of this study is the risk of bias introduced with a deliberative discussion focus group design. All focus group research should consider elements of reflexivity and trust amongst participants as they engage in dialogue [30]. While researchers discussed and considered these elements in the development of our interview guide, it is possible that some potential sources of bias were overlooked, and participants may have made assumptions unknown to the researchers. While we attempted to maintain objectivity and mitigate potential limitations, there is also the potential that the moderator and co-moderator may have projected bias onto the group.

Another limitation is that the input of patients and caregivers was not explored due to the focus on clinical provider perspectives. Future studies should conduct a more robust exploration, including dyadic patient-provider interaction research, and future work should also integrate triadic perspectives of providers, patients, and caregivers. Additionally, the small sample size limits the scope of perspectives and may not fully capture the breadth of provider experiences in rural and medically limited settings. This reflects both the scope of the project and the reality of provider shortages in rural and low medical access settings, where it is challenging to recruit larger numbers of clinicians to participate. While this size aligns with the intent of a formative phase aimed at generating preliminary, practice-informed insights, it limits the generalizability of these findings. Our intent was not to achieve statistical generalizability but rather to capture in-depth, practice-informed insights from providers directly engaged with healthcare delivery in these settings. Despite the small sample, the recurring themes observed across the information-rich group support that core perspectives were adequately represented for these objectives, suggesting adequate representation of key clinical experiences in these environments. We considered the emergence of consistent, repeated themes during the discussion. Finally, we only included providers in rural SC, thus limiting the generalizability to other geographic settings and provider populations. This exploratory phase serves as foundational work to inform subsequent implementation efforts.

Despite these limitations, the depth of provider insight offers valuable guidance for pilot testing remote spirometry integration within medically limited settings and set the foundation for future work in this area. Future work should expand to a broader range of providers and settings and must include patient perspectives and unmet needs. Larger studies are needed to examine clinical outcomes associated with remote spirometry and to explore strategies for data integration across EHR platforms. Incorporating patient and caregiver perspectives will be essential to provide a more holistic understanding of mHealth adoption and usability in real-world settings to ensure implementation strategies are responsive to the needs of all stakeholders.

## 5. Conclusions

Remote spirometry and mHealth, when supported by targeted implementation support and user-centered design, may serve as a critical bridge to COPD management in resource-limited settings. As the burden of COPD continues to grow, particularly in rural areas, innovative care delivery models are increasingly needed. These findings demonstrate that healthcare providers recognize the clinical value and potential of remote spirometry and mHealth tools to address diagnostic gaps, support self-management, and extend care for COPD. Successful implementation will require structured workflows, ongoing training, tailored data displays, and strong technical support. As digital health technologies continue to evolve, centering provider experience in their implementation is critical to achieving guideline-concordant, high-quality care.

This study represents an early step by identifying factors that must be addressed before scaling up remote spirometry in rural care. Building on these formative insights, future research should apply implementation science approaches to examine how remote spirometry and mHealth tools can be effectively integrated into existing clinical workflows. Priority areas include developing targeted training programs for clinicians and staff, conducting technical validation studies to ensure device accuracy and reliability in the real-world environment, and implementing pilot interventions to evaluate workflow feasibility, patient engagement, and clinical outcomes. Research should explore organizational, technological, and policy-level factors that influence adoption, sustainability, scale-up, and cost. Evaluating clinical outcomes such as exacerbation frequency, medication adherence, and hospitalization rates will demonstrate clinical impact and assess potential cost savings. Improvements in these areas may reduce avoidable healthcare utilization, which is especially important in resource-constrained settings where emergency care and hospitalizations represent significant financial strain. Finally, cost-effectiveness analyses are needed to identify implementation strategies that are financially sustainable, scalable, and capable of delivering equitable improvements in care.

## Figures and Tables

**Figure 1 nursrep-15-00402-f001:**
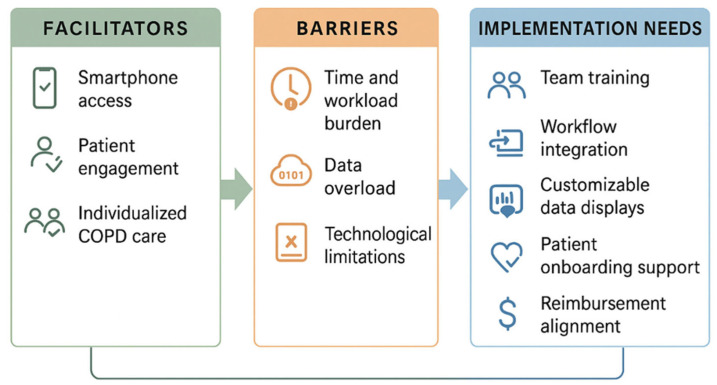
Provider-Identified Facilitators, Barriers and Implementations Needs for Remote Spirometry and mHealth in COPD Care.

**Table 1 nursrep-15-00402-t001:** mHealth Utilization Facilitators, Barriers and Implementation Needs.

Main Theme	Sub-Theme	Description	Participant Quotes
Facilitators	Addressing barriers in access to care	Accessibility and availability of mHealth are potential solutions to healthcare access barriers.	Participant 4, *“…we were surprised, actually, at how (pause) willing … to meet or used to telehealth technologies, … I think that’s … less and less of a barrier … it’s very rare that they don’t have a smartphone now.”* Participant 5, *“In addition to the limited resources, our patients often have limited capacity, whether that’s transportation or whatever …. Even if we can get the referrals and the [specialist] appointments set up …, they often can’t make that.”*
	COPD care (diagnostic and treatment support)	Remote spirometry reduces reliance on presumptive treatment and enhances decision-making.	Participant 4, *“… it’s a real struggle to try to predict which one it is. It’s hard for us clinically to make the right decisions when we don’t have the right information”.* Participant 5, *“And this technology would at least help us quickly rule out most pulmonary etiologies and help us triage because our patients have very limited resources.”*
	COPD self-management	Patients can be empowered through education, monitoring, and visual feedback to support adherence and shared decision-making.	Participant 5, *“… and you can show them in real time…it makes the conversation much easier. And I think for the patient, then they understand the rationale behind it.”*
2. Barriers	Time and workload	Adoption concerns for providers with limited time and increased workload.	Participant 5, *“In the middle of seeing a lot of patients … that’s just a lot and that’s probably not going to happen.”*
	Data volume and overload	Concerns about managing and interpreting the large amount of data; desire for concise, actionable summaries.	Participant 5, *“It’s got to be reliable and accurate … otherwise it’s just too much work.”* Participant 4, *“You want the summary stuff to just kind of be eye-popping.”*
	Technology barriers	Potential malfunctions (internet connectivity, device issues) and risk of reduced patient engagement during remote visits.	Participant 2, *“The patient’s there and they’re engaged but there’s a lot of distractions.”* Participant 5, *“If I was going to have glitchy things … it can be very difficult to have a good, meaningful conversation.”*
3. Implementation Needs	Clear processes, training, and support	Providers need guidelines, patient/staff training, technical support, data visualization, and clarity on reimbursement.	Participant 5, *“… to really have meaningful integration, there has to be support in that space for the provider: of getting the patient set up, getting them educated, getting them as well as myself connected …”* Participant 4, *“Are you gonna come up with…generalized guidelines …?”*

## Data Availability

Datasets are available from the PI upon request.

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
