# Peer review of "Clinical Provider Perspectives on Remote Spirometry and mHealth for COPD"

_nursrep, 2025, doi:10.3390/nursrep15110402_

Round 1

Reviewer 1 Report

Comments and Suggestions for Authors

This study focuses on the application of remote spirometry and mobile health (mHealth) in the management of chronic obstructive pulmonary disease in rural and medically underserved areas of the United States. By using the deliberative discussion focus group method to collect the perspectives of clinical providers, it fills the research gap regarding the preferences of clinical providers for the application of remote spirometry and mHealth tools in rural and low medical accessibility environments. The research design adheres to the norms of qualitative research, employing a semi-structured discussion guide and a six-step thematic analysis method. The conclusions are closely supported by data and have practical guiding value for the subsequent implementation of mHealth tools in the management of COPD at the grassroots level.

To further enhance the scientificity and practicality of the manuscript, it is suggested that the author make supplements or revisions in the following aspects:

  1. The study aimed to explore the views of clinical providers on mHealth with remote spirometry devices. However, based solely on the interview data of five participants, the sample size was insufficient to support the achievement of the research objective and was not convincing.
  2. The semi-structured interview guide merely mentioned that it was to facilitate discussions on mHealth, technology in clinical practice, and the care of COPD patients in medically restricted environments, lacking specific question examples.
  3. This study failed to clearly describe the background of the host and their training or experience in promoting deliberation (rather than leading to specific conclusions), which is not conducive to ensuring the objectivity of the interviews.
  4. The manuscript does not mention whether the research findings will be fed back to the participants for verification. Please improve.
  5. The format of the references is not uniform, and some are missing DOI links, which does not conform to academic norms and brings inconvenience to literature traceability. It is recommended to review and improve them in accordance with the requirements of the journal.
  6. The introduction section of this article does not reflect the necessity of spirometry devices for COPD patients. This part needs to be elaborated in detail in the article.
  7. The classification of themes and sub-themes is not clear; Secondly, there is a disconnection between some themes and the anchor sentences that support them, failing to form a consistent chain of argumentation. Please revise the theme framework and the corresponding relationship of quotations.

Author Response

Comments 1: (general comment) This study focuses on the application of remote spirometry and mobile health (mHealth) in the management of chronic obstructive pulmonary disease in rural and medically underserved areas of the United States. By using the deliberative discussion focus group method to collect the perspectives of clinical providers, it fills the research gap regarding the preferences of clinical providers for the application of remote spirometry and mHealth tools in rural and low medical accessibility environments. The research design adheres to the norms of qualitative research, employing a semi-structured discussion guide and a six-step thematic analysis method. The conclusions are closely supported by data and have practical guiding value for the subsequent implementation of mHealth tools in the management of COPD at the grassroots level. 

Response 1: We sincerely thank Reviewer 1 for their thoughtful and positive assessment of the study. We appreciate the recognition of our contribution to addressing a critical research gap by exploring provider perspectives on remote spirometry and mHealth implementation in rural and medically underserved settings. We are also grateful for the acknowledgment that our qualitative design, deliberative discussion approach, and thematic analysis align with established methodological standards and that our conclusions offer practical guidance for future implementation. In revising the manuscript, we have further strengthened its clarity and applied relevance by enhancing the Discussion section (page 9, section 4. Discussion, paragraph 1, line 353) to more explicitly link provider perspectives to real-world adoption challenges and by adding forward-looking recommendations for training, technical validation, and pilot interventions to support scale-up of remote spirometry in rural care. (page 13, section 5. Conclusion, paragraph 2, line 505)  

Comments 2: The study aimed to explore the views of clinical providers on mHealth with remote spirometry devices. However, based solely on the interview data of five participants, the sample size was insufficient to support the achievement of the research objective and was not convincing. 

Response 2: The authors agree with the reviewer’s point regarding sample size and regional focus. In response, we have expanded the Strengths and Limitations section to address this limitation explicitly. (page 11, section 4.1 Strengths and Limitations, paragraph 3, line 465) We now acknowledge that the small number of participants limits the breadth and generalizability of findings, consistent with the formative, exploratory intent of this qualitative phase. We clarify that this phase was designed to generate practice-informed insights from providers in medically limited settings to inform subsequent implementation research. (page 11, section 4.1 Strengths and Limitations, paragraph 3, line 471) Despite the small sample, recurring and information-rich themes were identified, suggesting adequate representation of core provider experiences. (page 11, section 4.1 Strengths and Limitations, paragraph 3, line 473) We also added forward-facing text specifying that these findings will inform a larger, multi-site, mixed-methods study evaluating feasibility, usability, and workflow integration of remote spirometry and mHealth tools in diverse rural contexts. (page 11, section 4.1 Strengths and Limitations, paragraph 3, line 479)  

Comments 3: The semi-structured interview guide merely mentioned that it was to facilitate discussions on mHealth, technology in clinical practice, and the care of COPD patients in medically restricted environments, lacking specific question examples. 

Response 3: The authors thank the reviewer for this helpful suggestion. We have revised the Study Design section to include illustrative examples from the semi-structured interview guide, clarifying the types of open-ended questions used to prompt discussion. These additions enhance transparency and reproducibility of the qualitative approach. (page 4, section  2.1 Study Design, paragraph 2, line 130) 

Comments 4: This study failed to clearly describe the background of the host and their training or experience in promoting deliberation (rather than leading to specific conclusions), which is not conducive to ensuring the objectivity of the interviews. 

Response 4: The authors appreciate this important point. The Procedure section has been expanded to clarify the moderators’ backgrounds and training, including their prior qualitative research experience and specific preparation to facilitate balanced deliberation and minimize bias. (page 4, section 2.2 Procedure, paragraph 1, line 139) 

Comments 5: The manuscript does not mention whether the research findings will be fed back to the participants for verification. Please improve. 

Response 5: The authors thank the reviewer for this valuable observation. While direct member checking was not conducted due to the quality improvement context and de-identified data collection, we have clarified this in the Data Analysis section. We now describe how analytic rigor was ensured through consensus coding, audit trail documentation, and iterative theme refinement among the research team. (page 4, section 2.3 Data Analysis, paragraph 1. line 146) 

Comments 6: The format of the references is not uniform, and some are missing DOI links, which does not conform to academic norms and brings inconvenience to literature traceability. It is recommended to review and improve them in accordance with the requirements of the journal. 

Response 6: The authors thank the reviewer for identifying these formatting inconsistencies. We have thoroughly reviewed and revised all references, ensuring uniform formatting and the inclusion of DOI links where available, in full accordance with the journal’s style requirements. (page 14, section References, line 555) 

 Comments 7: The introduction section of this article does not reflect the necessity of spirometry devices for COPD patients. This part needs to be elaborated in detail in the article. 

Response 7: We thank the reviewer for this valuable observation. The Introduction section has been substantially expanded to provide a more detailed rationale for the necessity of spirometry in COPD diagnosis and management. The revised text now explains that spirometry is the clinical “gold standard” for confirming COPD, differentiating it from other respiratory conditions, and guiding evidence-based treatment decisions according to GOLD (Global Initiative for Chronic Obstructive Lung Disease) guidelines. (page 2, section 1 Introduction, paragraph 2, line 61) We also discuss the potential clinical consequences of limited spirometry access in rural and medically underserved settings including underdiagnosis, misclassification, and delayed treatment. These revisions strengthen the rationale for remote spirometry integration in rural care.  

Comments 8: The classification of themes and sub-themes is not clear; Secondly, there is a disconnection between some themes and the anchor sentences that support them, failing to form a consistent chain of argumentation. Please revise the theme framework and the corresponding relationship of quotations. 

Response 8: The authors appreciate this insightful comment and the opportunity to examine the clarity and coherence of the thematic framework. In response, we carefully reviewed the classification of themes and sub-themes to ensure a consistent alignment between each theme and its supporting quotations. The framework maintains the overarching categories of barriers, facilitators, and implementation needs, which we determined to best represent the data and study objectives. Additionally, at the Academic Editor’s request, we created a visual diagram (Figure 1) to enhance clarity and cohesion, illustrating the interrelationships among barriers, facilitators, and implementation needs. (page 9, section 4 Discussion, paragraph 1, line 354) 

Reviewer 2 Report

Comments and Suggestions for Authors

General Comments

The manuscript reads well overall, though at times it feels a bit more like a policy summary than a clinical exploration. The discussion could go further in showing how these provider attitudes might translate into real-world adoption — or where they might not. Some sections repeat similar ideas, especially around workload and data management, which could be tightened.

Still, the strength of this work lies in the clarity of its main message: clinicians are open to innovation, but practical barriers — time, training, and infrastructure — continue to slow implementation. That is an honest and valuable finding.
Specific Comments

.- Study Design
The qualitative design using deliberative focus groups makes sense for the research question. Still, the authors could briefly explain how the participants were chosen and why the study was limited to South Carolina. That context helps the reader judge how far the findings can be generalized.

.- Participants
It would really help to see, at a glance, who took part. A short, simple table showing the professional roles, years in practice, and workplace setting would make the group easier to picture and give the reader a sense of the diversity of viewpoints.

.- Data Analysis
The authors describe the coding process carefully, maybe even a bit too carefully. This section could be tightened, focusing less on technical language and more on how consistency was ensured — for example, whether more than one researcher coded the data or if they checked each other’s work.

.- Results
The results are rich and the quotes are well chosen. However, Table 1 could be reorganized so that each theme is followed by one short “key takeaway” and a representative quote. That would make the findings easier to grasp and avoid repeating the same information in the text.

.- Discussion
The discussion should open with the main message: that clinicians are positive about remote spirometry but face very practical hurdles in daily care. It would be useful to link these observations to recent telehealth studies that looked at real-world implementation. Also, a gentle reminder that accepting an idea and being able to apply it are not the same thing. Finally, since the researchers are themselves clinicians, a brief reflection on how that dual role may have influenced interpretation would show transparency.

.- Tables and Figures
The text is detailed, but a single clear visual could do wonders — for example, a one-page diagram showing facilitators, barriers, and what participants said they would need to make the system work. Clinicians, administrators, and policymakers tend to absorb this type of summary much faster than long descriptive paragraphs.

.- References
The references are solid, but a few more from recent implementation or behavior-change literature would add depth. One or two key citations (for instance, Proctor’s framework on implementation outcomes or similar models) would anchor the discussion in a stronger theoretical base.

.- Conclusions
The conclusions are appropriate. I would encourage the authors to underline that this study represents an early step — identifying what needs to be addressed before scaling up remote spirometry in rural care. A short paragraph on next steps (training programs, technical validation, or pilot interventions) would make the paper feel forward-looking.

Author Response

Comment 1: (general comment) The manuscript reads well overall, though at times it feels a bit more like a policy summary than a clinical exploration. The discussion could go further in showing how these provider attitudes might translate into real-world adoption — or where they might not. Some sections repeat similar ideas, especially around workload and data management, which could be tightened. Still, the strength of this work lies in the clarity of its main message: clinicians are open to innovation, but practical barriers — time, training, and infrastructure — continue to slow implementation. That is an honest and valuable finding.

Response 1: We thank the reviewer for this thoughtful and encouraging feedback. In response, we refined the Discussion to better connect provider attitudes with the realities of real-world adoption, clarifying where enthusiasm may not translate into implementation without organizational readiness, training, and infrastructure support. We also tightened sections that previously reiterated concepts related to workload and data management to improve clarity and flow. These revisions strengthen the paper’s clinical focus while preserving the broader implications for policy and system-level change. We appreciate the reviewer’s recognition that the study’s strength lies in its honest portrayal of both clinician openness to innovation and the practical challenges that must be addressed to achieve sustainable adoption. 

Comment 2: Study design- The qualitative design using deliberative focus groups makes sense for the research question. Still, the authors could briefly explain how the participants were chosen and why the study was limited to South Carolina. That context helps the reader judge how far the findings can be generalized. 

Response 2: The authors appreciate this helpful suggestion. We have revised the Study Design to clarify participant selection and the geographic focus. (page 4, section  2.1 Study Design, paragraph 1, line 116) Specifically, we now note that participants were recruited using convenience and snowball sampling through professional and clinical networks to identify providers working in rural and medically limited settings. We also explain that South Carolina was selected because it has one of the highest burdens of COPD and among the lowest access to pulmonary specialty care in the U.S., making it an ideal context for exploring implementation challenges in resource-limited environments. These additions provide context to help readers assess transferability of findings. 

Comment 3: Participants- It would really help to see, at a glance, who took part. A short, simple table showing the professional roles, years in practice, and workplace setting would make the group easier to picture and give the reader a sense of the diversity of viewpoints. 

Response 3: The authors appreciate this helpful recommendation.  Given the small, regional sample, displaying each participants’  role, setting, and years in practice in a table may create concerns about participant confidentiality.  In response, we revised the results section summarize participant characteristics and quantify the number of participants in each professional role, the range of years in service, and workplace settings.   This addition provides readers with a clear overview of who participated and conveys the diversity of clinical perspectives represented in the focus group. (page 4, section  3 Results, paragraph 1, line 160)

Comment 4: Data Analysis- The authors describe the coding process carefully, maybe even a bit too carefully. This section could be tightened, focusing less on technical language and more on how consistency was ensured — for example, whether more than one researcher coded the data or if they checked each other’s work. 

Response 4: We thank the reviewer for this feedback. The Data Analysis section has been revised and streamlined, reducing the amount of technical detail while clarifying how analytic consistency was ensured. We now emphasize the iterative review process, including cross-checking of coding by multiple researchers and regular consensus meetings to ensure reliability and depth of interpretation. (page 4, section  2.3 Data Analysis, paragraph 1, line 147)

Comment 5: Results-The results are rich, and the quotes are well chosen. However, Table 1 could be reorganized so that each theme is followed by one short “key takeaway” and a representative quote. That would make the findings easier to grasp and avoid repeating the same information in the text. 

Response 5: The authors thank the reviewer for this constructive feedback. With the addition of Figure 1, we believe this now effectively offers a visual overview of the key takeaways and main findings without redundancy. (page 9, section  4. Discussion, paragraph 1, line 353) We opted to keep Table 1 for readers who prefer data presentation in this format for a quick glance at representative quotes and findings. (page 5, section  3.1. Themes, paragraph 1, line 178) We hope this allows readers to quickly understand the major findings while maintaining the richness of participants’ voices, and it complements rather than repeats the narrative description in the text. 

Comment 6: Discussion- The discussion should open with the main message: that clinicians are positive about remote spirometry but face very practical hurdles in daily care. It would be useful to link these observations to recent telehealth studies that looked at real-world implementation. Also, a gentle reminder that accepting an idea and being able to apply it are not the same thing. Finally, since the researchers are themselves clinicians, a brief reflection on how that dual role may have influenced interpretation would show transparency. 

Response 6: The authors appreciate this insightful comment. We have revised the Discussion section to more clearly foreground the study’s main message—that clinicians expressed overall optimism about the potential of remote spirometry and mHealth for COPD care but identified substantial practical barriers to real-world implementation. (page 9, section  4 Discussion, paragraph 1, line 356) The opening paragraph now explicitly contrasts provider acceptance with the challenges of application in daily practice, supported by references to recent telehealth implementation studies examining workflow integration, clinician workload, and sustainability in chronic disease management. We also added a reflexivity statement acknowledging that, as clinician-researchers, our dual roles may have influenced data interpretation. . (page 12, section  4.1 Strengths and Limitations, paragraph 2, line 455)  This addition enhances transparency and demonstrates how we mitigated potential bias through reflexivity, peer debriefing, and consensus coding. 

Comment 7: Tables and Figures- The text is detailed, but a single clear visual could do wonders — for example, a one-page diagram showing facilitators, barriers, and what participants said they would need to make the system work. Clinicians, administrators, and policymakers tend to absorb this type of summary much faster than long descriptive paragraphs. 

Response 7: We agree with the reviewer that a visual summary enhances accessibility for a broader audience. Accordingly, we have added a new Figure 1 illustrating the study’s key findings (facilitators, barriers, and implementation needs). This conceptual diagram provides a visual overview that complements the qualitative narrative, helping clinicians, administrators, and policymakers quickly grasp the practical implications of the results. We thank the reviewer for this suggestion. . (page 9, section  4 Discussion, paragraph 1, line 356)

 Comment 8: References- The references are solid, but a few more from recent implementation or behavior-change literature would add depth. One or two key citations (for instance, Proctor’s framework on implementation outcomes or similar models) would anchor the discussion in a stronger theoretical base.

Response 8: The authors thank the reviewer for their astute evaluation of references and recommendations to anchor the discussion in stronger theoretical base. Appropriate references discussing implementation outcomes, user-centered design and real-world implementation were added. (page 9, section  4 Discussion, paragraph 1, line 354)

Comment 9: The conclusions are appropriate. I would encourage the authors to underline that this study represents an early step—identifying what needs to be addressed before scaling up remote spirometry in rural care. A short paragraph on next steps (training programs, technical validation, or pilot interventions) would make the paper feel forward-looking. 

Response 9: We thank the reviewer for this excellent suggestion. We have revised the Conclusions section to explicitly highlight that this study represents an early, formative step in identifying key factors that must be addressed before scaling up remote spirometry in rural care. In response to this feedback, we added a forward-looking paragraph outlining specific next steps, including development of targeted provider training programs, technical validation studies to ensure device accuracy in real-world environments, and pilot implementation interventions to assess workflow feasibility, patient engagement, and clinical outcomes. These revisions strengthen the conclusion by positioning the study within a clear trajectory of ongoing implementation research aimed at achieving sustainable, scalable, and equitable integration of remote spirometry in rural and medically underserved settings. (page 11, section  5 Conclusions, paragraph 2, line 502)